# Recommendations for the Diagnosis and Treatment of Multiple Sclerosis Relapses

**DOI:** 10.3390/jpm12010006

**Published:** 2021-12-22

**Authors:** Cristina Ramo-Tello, Yolanda Blanco, Luis Brieva, Bonaventura Casanova, Eva Martínez-Cáceres, Daniel Ontaneda, Lluís Ramió-Torrentá, Àlex Rovira

**Affiliations:** 1Multiple Sclerosis and Clinical Neuroimmunology Unit, Germans Trias University Hospital, 08916 Badalona, Spain; 2Multiple Sclerosis Unit, Clínic Hospital, 08036 Barcelona, Spain; yblanco@clinic.ub.es; 3Multiple Sclerosis Unit, IRBLLEIDA. Arnau de Vilanova Hospital, 25198 Lleida, Spain; lbrieva.lleida.ics@gencat.cat; 4Multiple Sclerosis and Clinical Neuroimmunology Unit, La Fe Hospital, 46026 Valencia, Spain; Casanova_bon@gva.es; 5Immunology Service, LCMN, Germans Trias University Hospital, 08916 Badalona, Spain; evmcaceres@gmail.com; 6Mellen Center for Multiple Sclerosis, Cleveland Clinic, Cleveland, OH 44195, USA; ontaned@ccf.org; 7Multiple Sclerosis and Neuroimmunology Unit, Dr. Josep Trueta University Hospital and Santa Caterina Hospital, IDIBGI, 17004 Girona, Spain; llramio.girona.ics@gencat.cat; 8Department of Medical Sciences, University of Girona, 17004 Girona, Spain; 9Section of Neuroradiology, Radiology Service, Vall d’Hebron Universitary Hospital, 08035 Barcelona, Spain; alex.rovira.idi@gencat.cat

**Keywords:** multiple sclerosis, relapse, pseudo-relapses, methylprednisolone, treatment algorithm

## Abstract

Minimizing the risk of relapse is essential in multiple sclerosis (MS). As none of the treatments currently available are capable of completely preventing relapses, treatment of these episodes remains a cornerstone of MS care. The objective of this manuscript is to reduce uncertainty and improve quality of care of this neurological process. This article addresses definitions of key concepts, recommendations for clinical examination, classification criteria, magnetic resonance imaging, biomarkers, and specific therapeutic counsels including special populations such as pregnant and breastfeeding women, and children. An algorithm for treating MS relapses is also provided.

## 1. Introduction

Multiple sclerosis (MS) is an autoimmune disease of the central nervous system (CNS). It is characterized by inflammation (clinically expressed in the form of relapses), multifocal demyelination, axonal loss, and gliosis in both the white and gray matter. Currently, the cause of MS remains unknown. In experimental autoimmune encephalomyelitis (EAE), an animal model of MS, myelin-specific T cells are believed to play a crucial role in pathogenesis [1]. Much has been published on the role of disease-modifying therapies (DMTs) in reducing annualized relapse rates and relapse severity in MS, but there is comparatively less evidence, or consensus on how relapses should be diagnosed or treated. Early detection and optimal management of relapses will help ensure appropriate control of the disease. A panel of eight experts in the management of MS (the authors) —six neurologists, a neuroradiologist, and an immunologist—was formed to develop this report on the diagnosis and treatment of MS relapses. The aim was to provide a framework to help reduce variability in clinical practice.

## 2. General Principles

### 2.1. Relapse

A relapse is the consequence of an immune-mediated attack on the CNS. A patient is suspected of having a relapse when the person reports or is objectively observed to have evidence of a typical acute inflammatory demyelinating event in the central nervous system. Relapses are defined as clinical episodes lasting at least 24 h, in the absence of fever, infection or acute concurrent medical illness [2]. Relapses are also referred to as episodes, bouts, attacks, flares, flare-ups, or exacerbations.

The typical onset of a relapse in MS is subacute with intensity increasing over days, although cases of rapid onset have been described. In most cases, symptoms peak for about 1 to 2 weeks and then typically remit over the next 2 to 4 weeks without treatment. Based on the above time frames, for a relapse to be considered a second distinct event (second episode of CNS inflammatory activity), symptoms must occur at least 30 days after the start of the most recent flare and the new, recurring, or worsening symptoms must last for at least 24 h.

Agreement exists that the appearance of new symptoms within 30 days of initial onset corresponds to worsening of an existing episode, not a new episode. If the symptoms during this period are different to those at onset, the episode is considered to be multifocal. Other conditions, such as spinal cord compression, cerebrovascular disease, pseudo-relapses, and functional syndromes, must always be ruled out when a MS relapse is suspected.

### 2.2. Pseudo-Relapses

A pseudo-relapse is a clinical episode presenting with signs and symptoms similar to those observed in a previous relapse but in the absence of new inflammatory activity is due to systemic factors that can cause worsening of pre-existing neurologic symptoms. Its onset and resolution typically coincide with the triggering event. The possibility of relapse must be considered in patients whose symptoms persist after resolution of the triggering event or are more severe than prior episodes.

The main triggers of pseudo-relapses are infection (normally urinary or upper respiratory tract), stress, and increased body temperature due to external factors (e.g., a hot shower or high temperatures) or internal factors (e.g., fever or exertion) [3]. Worsening of symptoms caused by an increase in body temperature is known as Uhthoff’s phenomenon.

Misdiagnosis of pseudo-relapses often leads to inappropriate corticosteroid treatment or changes to DMT or immunosuppressive therapy, resulting in unnecessary risks and adverse effects.

### 2.3. Paroxysmal Symptoms

Paroxysmal symptoms are less common manifestations of MS. Paroxysmal symptoms are characterized by their sudden onset, brevity (usually seconds to a few minutes) [4], frequency (from 10 to 20 times per day up to a few hundred times per day), stereotyped fashion, relatively long clinical course (at least 24 h) and typically respond to carbamazepine [5].

Classic paroxysmal symptoms include trigeminal neuralgia, Lhermitte’s sign, clusters of tonic spasms, itching, paroxysmal diplopia, paroxysmal dysarthria with ataxia, paroxysmal paresthesia, and paroxysmal hemiparesis or hemitonic spasms. They are not accompanied by altered consciousness or changes in electroencephalographic activity.

When these symptoms occur in isolation, and particularly when they present for the first time, they may be the result of inflammatory activity indicative of a first relapse [6]. If they have occurred previously, they could reflect ephaptic transmission from an existing demyelinating plaque rather than acute disease activity. It is important to recognize and treat them with corticosteroids or carbamazepine as appropriate.

Epileptic seizures, aphasia, and other manifestations of cortical lesions are not considered paroxysmal symptoms in this context.

### 2.4. Relapse Triggers to Keep in Mind

Relapse rates are higher during the first three months postpartum [7]. Although the evidence is inconclusive, infections [8] including SARS-CoV-2 [9], vaccines [10] and stress [11] may trigger MS relapses too. Withdrawal of an effective DMT [12], monoclonal antibodies targeting tumor necrosis factor alpha (TNF-*α*) [13], gonadotropin-releasing hormone antagonists (used in the treatment of infertility, hormone-sensitive breast and prostate cancers, certain gynecological disorders and as part of hormone therapy in transgender patients) can also increase the risk of MS relapse [14].

### 2.5. Relapse Phenotypes

Relapse symptoms are readily identified when they are the result of acute inflammation in the optic nerve, spinal cord or brainstem/cerebellum.

There is increasing evidence that cognitive impairment (e.g., reduced performance at school or work) and psychiatric alterations could be due to an isolated cognitive relapse [15] or non-reactive depression [16]. Therefore it is recommended to quantify all new symptoms using appropriate tests or scales that permit follow-up and comparisons over time.

Relapse presentation tends to be monofocal, but patients can develop multifocal symptoms involving different functional systems as well.

### 2.6. Relapse Severity

There is currently no validated tool on how to assess the severity of a MS relapse. Physical exam is the most important tool for assessing the severity of MS relapse, the authors stress the importance of using the Expanded Disability Status Scale (EDSS) and recording it in the patient’s medical record.

It would be helpful to apply the criteria followed in some clinical trials according to which a relapse can be considered to be mild when there is an EDSS increase of less than 1 point, moderate when there is an increase of 1 to 2.5 points, and severe when there is an increase of 3 or more points [17]. If pre-relapse information is not available in the medical record, a score of 2 or higher in the visual, brainstem, or pyramidal functional systems is required for optic neuritis, myelitis, or brainstem relapse respectively. In the case of relapse in an uncertain location, a score of at least 2 points in the EDSS is necessary [18,19].

The authors consider that it is important to quantify the severity and apply the proposed therapeutic algorithm (Figure 1) until more evidence is available. In this way, the EDSS at the time of relapse can be compared with the previous EDSS allowing to quantify the increase in disability related to the relapse (EDSS score on relapse minus previous EDSS score).

### 2.7. Relapse Recovery

Remission is not a synonym of recovery. Recovery is considered to occur when the patient’s disability levels return to pre-relapse levels. Patients who do not recover pre-relapse function and disability levels (measured by EDSS) within 6 months of treatment are considered to usually have permanent sequelae.

Some authors have postulated that severe relapses and longer duration are associated with a greater risk of sequelae [20]. There is some evidence that recovery after 12 weeks of treatment with methylprednisolone (MP) is determined by pre-relapse EDSS scores rather than relapse severity, with lower remission rates observed in patients with a pre-relapse score of over 3.0 [21].

## 3. Clinical Examination

The author’s recommendations for the evaluation of patients in relapse are summarized in Table 1.

## 4. Treatment

The treatment goals for patients with a MS relapse are to shorten the duration and severity of the episode, relieve symptoms and increase recovery.

### 4.1. Which Types of Relapse Should Be Treated?

Corticosteroids relieve and shorten the duration of relapse symptoms but do not prevent sequelae and do not modify the disease course of MS over time [22]. It is therefore recommended not to treat mild relapses (increase in EDSS < 1 point), sensory relapses in particular, unless they have a significant impact on quality of life, and unless the potential benefits of treatment outweigh the potential adverse effects (AEs). Patients with remitting symptoms are often not treated.

### 4.2. Therapeutic Window

Patients should be made aware of the importance of informing their MS specialist team as soon as they suspect a relapse to ensure prompt evaluation, exclusion of a pseudo-relapse, and initiation of high-dose corticosteroids where appropriate.

### 4.3. Corticosteroid Treatment

The standard first-line treatment for relapses is corticosteroid therapy using high-doses of methylprednisolone (MP). The UK NICE (National Institute for Health and Care Excellence) guidelines [23] recommend administering at least 500 mg of oral MP every 24 h for 5 consecutive days. An alternative is to recommend 1 g of oral MP for 3 to 5 consecutive days [18,24,25] even for optic neuritis [26] given scientific evidence available to date.

The oral formulation offers greater convenience: enables treatment during weekends, prevents the patient from having to go to an outpatient clinic (a significant advantage during the SARS-COVID-19 pandemic) and results in considerable savings in terms of healthcare costs and prevents work productivity loss. As the vast majority of countries do not have access to oral commercial formulations of high-dose MP, hospital pharmacies need to prepare these formulations or request the manufacturers to produce tablets of various doses. Ideally, patients should be prescribed 500 mg tablets to enable the administration of 500-mg/24 h for 5 consecutive days or 1000 (500 + 500) mg/24 h for 3–5 consecutive days. The US Food and Drug Administration (FDA) or the European Medicines Agency (EMA) have not approved MP for oral use.

1 g of intravenous MP for 3 to 5 consecutive days can be administered if preferred by the physician or patient, in cases where oral MP has failed or is not tolerated, and if hospitalization is required due to severe symptoms, or to monitor other medical or psychological conditions such as diabetes and depression.

There is evidence suggesting that tapering doses of oral corticosteroids following high-dose administration does not offer benefits [27] and there is no significant risk of adrenal suppression.

Patients must be informed that they should expect to see an improvement within an average of 1 to 2 weeks of starting the first course of MP.

Patients with moderate relapses who do not tolerate MP could be candidates for adrenocorticotropic hormone therapy [28] administered as an intramuscular or subcutaneous injection at 80–120 units for 14–21 days. A growing number of studies report favorable results in this setting. However, ACTH gel is significantly more expensive than steroids.

### 4.4. Inadequate Response to MP

The authors agreed that the definition of inadequate treatment response included worsening of the EDSS within 2 weeks of MP initiation. If the increase in EDSS is moderate (1–2.5 points), a second course of MP should be given as per the recommendations in the section on corticosteroid treatment. If the increase is substantial (≥3 points), treatment with plasmapheresis should contemplated. Plasmapheresis is the only second-line treatment for steroid-resistant exacerbations supported by solid clinical evidence [29]. The procedure is performed every other day to reach a total of five sessions, although a shorter may be used in patients showing an optimal response.

Patients with a suboptimal response after five plasmapheresis exchanges or patients in whom MRI shows persistent acute inflammatory activity should be started on a third-line therapy. Options include (1) cytotoxic therapy with myeloablative agents such as cyclophosphamide (200 mg/kg/d for 4 days intravenously) [30], or (2) an initial high loading dose of rituximab (2000 mg intravenously) divided into 2 infusions given 2 weeks apart to attack antibody-secreting B lymphocytes [31], or (3) a single dose of natalizumab (300 mg intravenously) to prevent lymphocytes from crossing the blood-brain barrier [32]. These third-line treatments, which can also be used for fulminant or tumefactive demyelination, are not supported by strong evidence, but are commonly used in severe cases.

Data suggest that intravenous immunoglobulin therapy is not useful [32,33,34], but it could be a suitable option for patients with severe relapses who do not have access to a center with plasmapheresis.

Patients with severe relapses who are intolerant to MP should be treated with plasmapheresis.

### 4.5. Adverse Effects of Methylprednisolone

Patients must be warned that they may experience immediate adverse effects (AEs) during treatment with high-dose MP. AEs are generally mild or moderate. The most common AEs are insomnia, mood changes (irritability, euphoria and depression), gastrointestinal disorders, palpitations, weight gain, edema, acne, headache, musculoskeletal pain, and a metallic taste in the mouth. Patients should also be warned about the risk of infections, such as herpes, sepsis, pneumonia, arthritis, bursitis, and complicated urinary infections.

The need for prophylactic treatment of AEs should be assessed on a case-by-case basis. Examples are insomnia medication and gastroprotective drugs for patients with a history of peptic ulcers, acid reflux, or hiatal hernia. Diabetic patients should be advised to monitor their blood sugar levels closely, and despite the lack of evidence, non-diabetic patients should undergo capillary blood glucose testing at their primary care center or local pharmacy. Blood pressure should also be monitored in patients with hypertension.

Serious events are rare. The most common serious event are psychosis, depression, and mania. There have been very few reports of reactivation of latent infections to date, but caution is warranted as patients on immunosuppressive DMT are at an increased risk. There are no data showing that high-dose MP significantly increases the risk of latent chronic infections.

High doses of MP are considered to be immunosuppressive. MP should not be administered 8 weeks before or 8 weeks after the administration of live or attenuated live vaccines because of the increased risk of complications due to viral replication [35]. Vaccines containing inactivated or killed organisms may be used, although response may be diminished.

### 4.6. Symptomatic Treatment during MS Relapses

It should be contemplated for pain, spasticity, and diminished sphincter control, as it can improve function, patient comfort, and quality of life.

The benefits of rehabilitation during acute relapse are not well established [36] but in general is recommended. Patients with fatigue and motor or sensory cord deficits may need to rest and be warned about the risk of falls, while those with sensory deficits for pain should be warned of the risk of injury.

### 4.7. Relapse Treatment during Pregnancy

The FDA does not classify MP as a pregnancy risk drug.

High-dose corticosteroids administered for short periods (3–5 days) appear to be safe during pregnancy, especially in the second and third trimesters.

There is conflicting evidence on the association between corticosteroid use during pregnancy and the risk of cleft lip and/or cleft palate [37]. There is little evidence that systemic corticosteroid use in pregnancy independently increases the risk of preterm birth, low birth weight, or preeclampsia. Currently, there is not enough evidence to determine whether systemic corticosteroids might contribute to gestational diabetes mellitus [38].

Intravenous immunoglobulins could be a suitable option for patients with severe relapses who do not respond to MP, although the risk of AEs such as cerebral venous thrombosis (which has an increased risk of occurring during pregnancy) must be contemplated [39]. Plasmapheresis performed under expert supervision can also be considered in such cases. The couple must always be involved in treatment decisions.

### 4.8. Relapse Treatment during Breastfeeding

Women with MS are at an increased risk of relapse in the postpartum period. Although the level of MP transfer into breast milk is very low, generally IVMP treatment is not advised during breastfeeding. If the mother wishes to continue breastfeeding, intravenous MP is preferable to oral, as it has a shorter peak effect. Patients should wait for 2–4 h before breastfeeding after administration of MP, as there is evidence that levels in infants after this time are lower than those in infants being treated with MP for another condition (0.25 mg/kg) [40]. Another option is to breastfeed only before receiving high doses of MP. Mothers can also use the approach of expressing milk using a breast pump, which can later be offered to the baby, before receiving MP.

### 4.9. Relapse Treatment in Children

Considering the lack of evidence regarding the treatment of MS relapses in the pediatric population, the recommendations in children are based on data from adult studies. MP is the most commonly selected first-line therapy for disabling MS relapses in children. The recommended treatment is intravenous MP 30 mg/kg/d (maximum 1000 mg/d) for 3–5 consecutive days [41]. Although equivalent studies have not been performed in children, it is assumed that high-dose oral corticosteroids will be similarly effective for the treatment of MS relapses in children and adults. Oral corticosteroid taper for the treatment of relapses is not a routine practice in pediatric MS. One exception is acute disseminated encephalomyelitis, where the recommended treatment is high-dose corticosteroids followed by an oral taper over 4–6 weeks with a starting dose of prednisone of 1–2 mg/kg/day. A taper period of 4–6 weeks is recommended as an increased risk of relapse has been observed with periods of 3 weeks or less [42].

As with adults, non-responders should be administered a second course of MP or treated with plasmapheresis [43].

The author’s algorithm for treating MS relapses is summarized in Figure 1.

### 4.10. Treatment of Asymptomatic Active Lesions on MRI

It is difficult to evaluate the immediate or delayed effects of treatment initiated for acute lesions without clinical correlates. Magnetic resonance Imaging (MRI) studies have shown that focal signal abnormalities reflecting inflammatory activity can precede clinical manifestations by several weeks [44]. Because of this mismatch between imaging and clinical manifestations, it is possible that by the time a patient develops symptoms and is prescribed corticosteroid treatment, he or she may already have developed permanent residual disability due to demyelination and even axonal damage. There are neurologists who think that an asymptomatic new lesion on MRI should be treated accordingly [45] and others who think otherwise, as there is no evidence of the comparative effectiveness of these approaches [46,47].

There is also no evidence available on whether or not to treat large or numerous asymptomatic lesions involving the brainstem or spinal cord.

The authors recommend performing a more exhaustive examination (using the SDMT, the Nine-Hole Peg Test, or a fatigue scale) in patients with “asymptomatic” active lesions detected by MRI to establish whether lesions are truly subclinical.

## 5. MRI Studies during Relapses

MRI is the gold standard diagnostic test for MS but it is often impractical to obtain in a timely fashion in relation to relapse and can miss lesions for determined variety of reasons. Brain and spinal cord MRI scans are not necessary in patients with a clear diagnosis of MS relapse. They are, however, recommended in patients with an unclear or non-objectively observed clinical relapse and should be performed before starting corticosteroid treatment, as corticosteroids can reduce the time during which lesions show contrast uptake. MRI should be performed ideally within the first 72 h if steroid treatment is planned, and within 2 to 3 weeks of relapse onset in patients that do not receive steroid treatment because mean enhancement duration without treatment is 2–4 weeks [48].

A brain MRI study can demonstrate gadolinium-enhancing lesions when cognitive impairment, depression, or even excessive fatigue is suspected to be due to a new episode of inflammatory activity.

Sometimes a spinal cord MRI is deemed necessary to confirm a relapse likely related to spinal cord involvement. A brain MRI is also necessary in such cases, as active spinal lesions tend to be associated with active cerebral lesions, which are easier to identify. This combined strategy increases the likelihood of detecting active inflammatory lesions. However, adding spinal cord MRI to brain MRI in patients with relapses related to brain lesions does not appear to increase the sensitivity of MRI for detecting disease activity.

The abbreviated MAGNIMS (Magnetic Resonance Imaging in MS) MRI protocol should be followed at all times [49,50]. The contrast agent should be injected at least 5, and ideally 10, minutes before obtaining the T1 sequence. This wait time can be used to obtain other sequences included in the protocol (T2-FLAIR, T2).

MRI studies, particularly MRI conducted with contrast, should be avoided during the first trimester of pregnancy.

The author’s recommendations for the MRI assessment of patients in relapse are summarized in Table 2.

## 6. Biomarkers

There have been reports of changes in lymphocyte subpopulations that can predict or support the occurrence of a relapse and that normalize after corticosteroid treatment [51,52]. There are, however, no immune related biomarkers currently available outside research applications.

Neurofilament light (NfL) chain protein is a cytoskeletal protein located in neuronal axons. Increased cerebrospinal fluid and blood NfL levels have been described in MS (following axonal injury) and other diseases. They have also been linked to aging. The possible value of NfL as a biomarker for MS relapses has been postulated in recent years, as increased levels have been associated with the presence of gadolinium-enhancing lesions on MRI [53]. In this context, in patients with early MS, the presence of both abnormal NfL and thin ganglion cell and inner plexiform layer in retinal optical coherence tomography (OCT) have been described as additive risk factors of disease activity [54].

More recently, wide metabolomic studies have shown metabolic perturbations during relapses, and several serum metabolites, mainly lysine and asparagine (higher in relapses), as well as isoleucine and leucine (lower in relapses), postulated as potential biomarkers useful to differentiate relapse from remission. Future metabolomics studies will need to prospectively include MRI scans to understand metabolic signatures and their relation with MRI-defined inflammation [55].

## 7. Conclusions

Diagnosis and particularly treatment of MS relapses varies greatly among clinicians. This document offers a series of simple and practical statements with recommendations that, although not reaching full consensus,, reflect the realities of current clinical practice. They are easy to implement in daily practice and can be readily adapted to the specific needs of practitioners seeking to standardize care processes in MS. Further studies on larger cohorts are required to confirm the effectiveness of the interventions regarding relapses in MS.

## Figures and Tables

**Figure 1 jpm-12-00006-f001:**
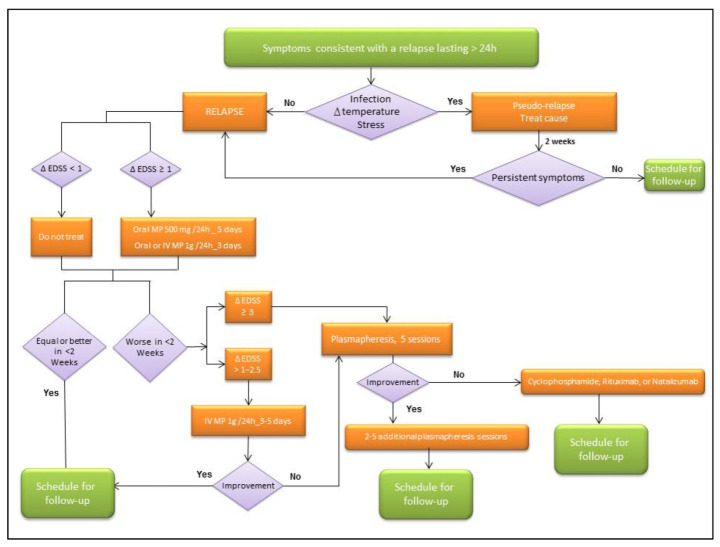
Therapeutic algorithm of MS relapse.

**Table 1 jpm-12-00006-t001:** Recommendations for the Assessment of Patients with Suspected Relapse.

Recommendations for the Assessment of Patients with Suspected Relapse
Mandatory:
- Date of onset Clinical topography and if monofocal or multifocal Visual acuity measured using a Snellen chart Visual contrast sensitivity test in patients with optic neuritis Functional system scores EDSS score Re-evaluate patients by telephone or electronic communication after 2 weeks and instruct them to contact a provider if they notice worsening of symptoms Follow-up visit at 6 months
Optional: - Symbol Digit Modalities Test (SDMT) Nine-hole peg test (9HPT) Timed 25-foot walk test (T25FW) (7.62 m)

**Table 2 jpm-12-00006-t002:** Recommendations for the MRI Assessment of Patients with Suspected Relapse.

Recommendations for the MRI Assessment of Patients with Suspected Relapse
- Obtain MRI for diagnostic and prognostic purposes during the first relapse (clinically isolated syndrome) Obtain MRI before treatment escalation in patients who do not respond to corticosteroids Obtain MRI before treatment initiation with corticosteroids in patients with severe, unexpected relapses Obtain MRI in patients with an unclear or non-objectively observed clinical diagnosis Obtain MRI in patients in whom gadolinium-enhanced lesions must be demonstrated before initiation of immunomodulatory treatment.

## Data Availability

Not applicable.

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
