# Peer review of "Recommendations for the Diagnosis and Treatment of Multiple Sclerosis Relapses"

_jpm, 2021, doi:10.3390/jpm12010006_

Round 1

Reviewer 1 Report

Overall, the subject is very interesting: standardization of diagnosis and treatment of multiple sclerosis is of great value in clinical practice. Accurate determination of relapses in multiple sclerosis is important for diagnosis, classification of clinical course and therapeutic decision making. I have several issues with the work presented, but am otherwise enthusiastic about the presentation, flow and coverage of the manuscript:

Major issue

i) As this is a journal dedicated to personalized medicine, I would expect the authors to expand section 6 so that it includes current and future perspective on use of relapse biomarkers, for example in Lin et al. (DOI: https://doi.org/10.1212/NXI.0000000000001051) and Yeo et al. (DOI:10.1093/braincomms/fcab240).

Minor issues

i) Page 3, line 98: “… brainstem/ cerebellum …“ should read “… brainstem/cerebellum …” (i.e. skip the space)

ii) Page 3, line 108: Rephrase into something like: “Since nowadays physical exam is …” Also, put a dot at the end of the sentence.

iii) Page 3, line 111: Rephrase into something like: “It would be helpful to apply…” Also, put a dot at the end of the sentence.

iv) Page 3, Table 1: Move Table 1 to page 4 for clarity reasons. Also, change the table title: Recommendations for the assessment of patients in relapse

v) Page 8, line 277: Re-format so that it reads “Figure 1.” and add the figure title and a short description.

vi) Page 8, Table 2: Re-format and rephrase the table as it does not make sense currently. Make the table more informative. Also add the title above the table so that it reads: “Recommendations for the MRI assessment of patients in relapse”.

Author Response

Dear referee,

Thank you very much for the interest you have taken in reading our work. 

We have commented on the text and included in the bibliography the interesting references you have provided to us. 

In this context, in patients with early MS, the presence of both abnormal NfL and thin ganglion cell and inner plexiform layer in retinal optical coherence tomography (OCT) have been described as additive risk factors of disease activity [53].

More recently, wide metabolomic studies have shown metabolic perturbations during relapses, and several serum metabolites, mainly lysine and asparagine (higher in relapses), as well as isoleucine and leucine (lower in relapses), postulated as potential biomarkers useful to differentiate relapse from remission. Future metabolomics studies will need to prospectively include MRI scans to understand metabolic signatures and their relation with MRI-defined inflammation [54].

We have also corrected the typographical errors, rephrased the tables and figure 1, and clarified the headings of all of them.

We have also had sent the text to be revised by a professional english translator.

Thank you very much again 

Reviewer 2 Report

Very comprehensive work!

Author Response

Dear reviewer,

We are delighted that you found our work interesting. We will proceed to revise the English language in the final version, as advised by you.

Thank you very much

Reviewer 3 Report

In this article, a panel of eight experts in the management of MS made a report on the diagnosis and treatment of MS relapses, aiming to provide a framework to help reduce inter-neurologist variability in these clinical process. They provide recommendations for clinical examination, classification criteria, magnetic resonance imaging, biomarkers, and specific therapeutic counsels including special populations such as pregnant and breastfeeding women, and children. The manuscript is straightforward, well written, and concise and has clear results within the scope of a review paper. Definitely deserves to be published and is a valuable contribution to the “Journal of Personalized Medicine”. Some minor flaws need to be addressed before publication.

Minor points:

[1] “Relapse triggers to keep in mind”, Lines 89-95:

Withdrawal of an effective DMT [10], monoclonal antibodies targeting tumor necrosis factor alpha (TNF-α) [11], gonadotropin-releasing hormone antagonists (used in the treatment of infertility, hormone-sensitive breast and prostate cancers, certain gynecological disorders and as part of hormone therapy in transgender patients) can also increase the risk of MS relapse [12].”.

At that point, the authors should incorporate atalizumab, a humanized monoclonal antibody (mAb) directed against the very late antigen 4 (VLA-4) adhesion complex. Atalizumab has been introduced into the treatment of MS and reduces the rate of clinical relapse in patients with relapsing MS.

Recommended reference: Kyritsis AP, et al. Cancer specific risk in multiple sclerosis patients. Crit Rev Oncol Hematol. 2016 Feb;98:29-34.

[2] General comment:

Is there any correlation between COVID-19 infection and multiple sclerosis relapses? In a recently published retrospective study, there was no increased risk of relapse, shortly following infection. However, should there be a long-term latent SARS-CoV-2 infection in the CNS parenchyma, its reactivation over long periods of time may hypothetically cause neurodegeneration and disease progression in MS patients. I encourage the authors to make a comment on that.

Recommended reference: Etemadifar M, et al. COVID-19 and the Risk of Relapse in Multiple Sclerosis Patients: A Fight with No Bystander Effect? Mult Scler Relat Disord. 2021 Jun;51:102915.

Author Response

Thank you very much for your comments.

In relation to natalizumab we consider that it is included in the paragraph "Withdrawal of an effective DMT [11] ,.... can also increase the risk of MS relapse [13]", because natalizumab is a DMT.

We have made specific mention of SARS-COVID-19 infection, as you indicate:  

Relapse rates are higher during the first three months postpartum [7]. Although the evidence is inconclusive, infections [8] including SARS-CoV-2 [9] , vaccines [10] and stress [11] may trigger MS relapses too. 

However, we believe that to talk about that a long-term latent SARS-CoV-2 infection in the CNS parenchyma and its reactivation over long periods of time may hypothetically cause neurodegeneration and disease progression in MS patients, is beyond the scope of this paper

Thank yoy very much again 

Reviewer 4 Report

The article is interesting a present a really good point of view. 

Minor changes: 

I would recommend authors to talk about immune system which is deeply involved in MS. PMID: 34440933

Author Response

Thank you very much for your comment. We have added the immunological definition of MS in the introduction.

Multiple sclerosis (MS) is an autoimmune disease of the central nervous system (CNS). It is characterized by inflammation (clinically expressed in the form of a relapse), multifocal demyelination, axonal loss, and gliosis in both the white and gray matter. Currently, the cause of MS remains unknown. In experimental autoimmune encephalomyelitis (EAE), an animal model of MS, myelin-specific T cells are believed to play a crucial role in its pathogenesis.